# A Temporary Pause in the Replication Licensing Restriction Leads to Rereplication during Early Human Cell Differentiation

**DOI:** 10.3390/cells11061060

**Published:** 2022-03-21

**Authors:** Marie Minet, Masood Abu-Halima, Yiqing Du, Julia Doerr, Christina Isted, Nicole Ludwig, Andreas Keller, Eckart Meese, Ulrike Fischer

**Affiliations:** 1Institute of Human Genetics, Saarland University, 66421 Homburg, Germany; marie.minet@uni-saarland.de (M.M.); masood.abu-halima@uks.eu (M.A.-H.); yiqingdu@outlook.com (Y.D.); julia.doerr67@gmail.com (J.D.); c.isted@gmx.de (C.I.); n.ludwig@mx.uni-saarland.de (N.L.); eckart.meese@uks.eu (E.M.); 2Chair for Clinical Bioinformatics, Saarland University, 66123 Saarbrücken, Germany; andreas.keller@ccb.uni-saarland.de

**Keywords:** gene amplification, fiber combing, CDT1, GMNN

## Abstract

Gene amplifications in amphibians and flies are known to occur during development and have been well characterized, unlike in mammalian cells, where they are predominantly investigated as an attribute of tumors. Recently, we first described gene amplifications in human and mouse neural stem cells, myoblasts, and mesenchymal stem cells during differentiation. The mechanism leading to gene amplifications in amphibians and flies depends on endocycles and multiple origin-firings. So far, there is no knowledge about a comparable mechanism in normal human cells. Here, we describe rereplication during the early myotube differentiation of human skeletal myoblast cells, using fiber combing and pulse-treatment with EdU (5′-Ethynyl-2′-deoxyuridine)/CldU (5-Chlor-2′-deoxyuridine) and IdU (5-Iodo-2′-deoxyuridine)/CldU. We found rereplication during a restricted time window between 2 h and 8 h after differentiation induction. Rereplication was detected in cells simultaneously with the amplification of the *MDM2* gene. Our findings support rereplication as a mechanism enabling gene amplification in normal human cells.

## 1. Introduction

DNA sequence amplification is a phenomenon that occurs predictably at defined stages during normal development in *Xenopus*, *Drosophila*, *Sciara,* and *Tetrahymena* [1,2,3,4]. These amplifications affect specific DNA regions and appear during narrow windows of development [4]. In normal mammalian cells, gene amplification remained disregarded for many years.

We published the first evidence of gene amplifications during differentiation in human neural progenitor cells [5], mouse neural stem and progenitor cells [6], mouse and human myoblasts [7], and mesenchymal stem cells [8]. Additional reports were published on gene amplification during human and mouse trophoblast differentiations [9,10]. So far, the mechanism for amplification, or the initiation of gene amplification, has yet to be elucidated in differentiating cells. Here, we set out to investigate whether rereplication is a possible mechanism leading to gene amplifications in human myoblasts following differentiation induction. It is likely that mammalian gene amplification during differentiation originates from mechanisms that have been described for chorion gene amplification in *Drosophila* eggshell cells. Reports of multiple origin-firings were published by Osheim YN as early as 1988 [11], and Alexander JL in 2015 [12].

The replication factors CDT1 and CDC6 are essential for origin licensing and replication initiation. There are several mechanisms that ensure the initiation of origins only once per cell cycle, including the degradation of CDT1 and CDC6, as well as CDT1 inhibition by geminin (GMNN) [13,14]. A recent study on transgenic mouse embryo stem cells with inducible *CDC6* and *CDT1* expressions revealed rereplication when *CDT1* and *CDC6* were simultaneously overexpressed. Here, we analyze *CDC6*, *CDT1,* and *GMNN* expressions in differentiating human skeletal myoblasts (HSkM) cells during a time window, where our studies detected rereplication by fiber combing.

## 2. Materials and Methods

### 2.1. Cell Culture and Differentiation

HSkM-S cells derived from primary normal human skeletal myoblasts were obtained from Thermo Fischer Scientific (Life Technologies, Carlsbad, CA, USA). For proliferation, HSkM cells were cultivated in Dulbecco’s Modified Eagle Medium (DMEM) (Life Technologies, Carlsbad, CA, USA), supplemented by 10% FCS (Fetal calf serum) (Biochrom, Berlin, Germany). For differentiation, the cells were cultivated for 24 h in Dulbecco’s Modified Eagle Medium (DMEM) supplemented by 2% Horse Serum (PAN Biotech, Aidenbach, Germany), according to the manufacturers’ instructions.

### 2.2. Thymidine Analogue Treatment

Cells were cultivated as described above. The media, that either supported proliferation or differentiation, were supplemented by thymidine analogues in two sequential pulses. A 90 min-long pulse with the first thymidine analogue was followed by a brief PBS wash step and a 15 min incubation in a medium without thymidine analogues, and was subsequently followed by a second 45 min pulse with a second, different, thymidine analogue. The thymidine analogue EdU (Life Technoligies, Carlsbad, CA, USA) was added to create a final concentration of 200 μM, CldU (Merck, Darmstadt, Germany) to create a final concentration of 250 µM, and IdU (Merck, Darmstadt, Germany) to create a final concentration of 30 µM. After all pulse labeling steps, the cells were harvested using Accutase, resuspended in PBS, and processed further.

### 2.3. Fiber Preparation for Molecular Combing

Harvested cells from the culture were diluted to 0.25 million cells/mL in PBS. The cells were then suspended in an agarose plug and allowed to digest overnight in an ESP buffer (containing proteinase K) at 50 °C. The plugs were then washed in a 1x TE buffer on an overhead rotator, and were subsequently digested overnight with agarase. The resulting DNA suspension was transferred to a combing reservoir obtained from the FiberPrep Kit (Genomic Vision, Bagneux, France) with a 0.1 M MES buffer at a pH of 6.5. The final DNA suspension in the FiberComb reservoirs were inserted into the FiberComb Molecular Combing Device, along with 1–2 silanized coverslips (Genomic Vision, Bagneux, France) per reservoir. The coverslips were then mechanically inserted into the DNA suspension and pulled out at a constant rate. This resulted in DNA fibers adhering to the silanized surface with a constant measurement of 2 kb/μm. Once the DNA was combed onto the coverslips, they were baked at 65 °C for two hours to fix the DNA onto the coverslip. They were then stored at −20 °C, or were processed further.

### 2.4. EdU Click-iT Reaction

Coverslips that were baked to fix the DNA were processed as described in the manual provided by Invitrogen™ (Thermo Fischer Scientific, Waltham, MA, USA). In brief, the coverslips were washed with 3% BSA and treated with the Click-iT™ reaction cocktail (containing Alexa-594), as detailed by Invitrogen™. This light-sensitive reaction was incubated for 30 min in a dark humid chamber at room temperature. After incubation, the coverslips were washed again in 3% BSA (Merck, Darmstadt, Germany). After washing, the solution was completely removed, and antibody immunostaining followed, as described below.

### 2.5. Fluorescent Immunostaining of Combed Fibers

Coverslips that were treated with the Click iT™ Imaging kit were then placed in a humid chamber with 10% goat serum diluted in PBS, and blocked for 30 min at 37 °C. This was followed by the rat anti BrdU (5-Bromo-2′-deoxyuridine) antibody for CldU (Abcam, Cambridge, UK), diluted 1:12.5 in 10% goat serum, and incubated for one hour in a humid chamber at 37 °C. The coverslips were subsequently washed in PBS, three times for 5 min each. Then, the corresponding secondary antibody solution (Alexa-488), diluted in 10% goat serum, was placed on the coverslips, which were incubated for 30 min in a humid chamber at room temperature. The coverslips were then washed again, dehydrated with an ethanol series, and mounted using SlowFade™ Gold (Invitrogen™, Thermo Fischer Scientific, Waltham, MA, USA). All coverslips were individually analyzed using the Olympus AX70 fluorescent microscope and the cellSens Dimension 1.8.1 Program from Olympus (Tokyo, Japan).

Cells with IdU/CldU pulse treatments for EasyComb (Genomic Vision) were harvested and embedded into agarose plugs using the Genomic Vision FiberPrep^®^ Kit (Genomic Vision, Bagneux, France). Combing and immunostaining were performed according to the EasyComb procedures (Genomic Vision, Bagneux, France). Coverslips were scanned with the FiberVision^®^ scanner and images were analyzed using Genomic Vision FiberStudio^®^ software (Genomic Vision, Bagneux, France). Intact IdU (green) and CldU (red) replication tracks, flanked by counterstaining (blue), were selected and used for further validation.

### 2.6. Fluorescence In Situ Hybridization

The BAC clone RP11-611O2 (*MDM2* gene) is from the RP-11 (http://www.chori.org/bacpac/, accessed on 1 October 2011) libraries of the Welcome Trust Sanger Institute and are available from SourceBioSciences, Germany. BAC-DNA (1 µg) was labeled with alexaFluor 488-dUTP (green fluorescence signals) using the FISH Tag DNA labeling Kit, according to the manufacturer’s instructions (Thermo Fischer Scientific, Waltham, MA, USA). Labeled probe DNA was precipitated in the presence of human Cot-1 DNA and resuspended in a hybridization mix (50% formamide, 20× SSPE, 20% dextrane sulfate, and 27% SDS).

HSkM cells were grown on glass slides and allowed to differentiate for 5 h. EdU (200 µM) was added for 45 min, and cells were briefly washed with PBS and supplied with a fresh differentiation medium for an additional 19 h. In addition, cells were differentiated for 2 h. Three hours after the differentiation induction, EdU was added for 3 h. Subsequently, cells were briefly washed with PBS and were supplied with a fresh differentiation medium for the remaining differentiation time (18 h). As a control experiment, HSkM cells were also cultured for 24 h in DMEM/10% FCS. Three hours after the proliferation start, EdU was added for 3 h. Again, cells were briefly washed with PBS and were supplied with a fresh DMEM/10% FCS medium for the remaining 18 h. Cells on glass slides were fixed in ice cold methanol for 15 min, washed in PBS for 5 min, and treated with 0.02% Tween-20 for 5 min. Slides were RNase-treated for 0.5 h at 37 °C and pepsin-treated for 10 min. Post-fixation was performed using 2.5% formaldehyde/1× PBS for 10 min at room temperature. Hybridization was performed over-night. After the post-hybridization washes, slides were treated according to the EdU Click-iT Reaction, as described above. Fluorescence images were captured with an Olympus AX70 microscope using cellSens software from Olympus.

### 2.7. RNA Isolation and qRT-PCR

The total RNA from HSkM cells during 24 h of differentiation were isolated using the QIAGEN miRNeasy Mini Kit (QIAGEN, Hilden, Germany) after 3 h, 6 h, 9 h, 12 h, and 24 h, during the 24 h differentiation course. Undifferentiated HSkM cells served as a 0 h control. mRNA was converted to cDNA using the QuantiTect RT Kit (Qiagen GmbH, Hilden, Germany). qRT-PCR was performed using the SYBR^®^ Green PCR Kit using 5 ng cDNA for qRT-PCR with QuantiTect Primer Assays. The Qiagen QuantiTect primer pairs are as follows: *GAPDH* (QT00079247), *TBP* (QT00000721), *CDT1* (QT00020601), *CDC6* (QT00065772), and *GMNN* (QT01019970). qRT-PCR was run on a StepOne™ Real-Time PCR System (Applied Biosystems™, Foster City, CA, USA). The relative expression was calculated with a ΔΔCt method with the endogenous controls *GAPDH* and *TBP*.

### 2.8. Protein Isolation and Western Blot Analysis

HSkM cells were harvested with Accutase, pelleted, and washed once with PBS. The pellet was resuspended in a RIPA buffer (Thermo Fisher Scientific) supplemented with a protease inhibitor (Sigma-Aldrich, St. Louis, MO, USA). The samples were sonicated at 20 joules for 2 s, incubated on ice for 30 min, and centrifuged at 14,000× *g* for 10 min. The supernatant was transferred to a new microcentrifuge tube. Twenty µg of total protein were denatured with a Laemmli buffer, separated by gel electrophoresis (SDS-PAGE) in a Mini-Protean^®^ TGX Precast Gel (Bio-Rad Laboratories, Hercules, CA, USA), and subsequently transferred onto polyvinylidene difluoride (PVDF) membranes (Whatman, Maidstone, UK). Membranes/blots were blocked for 1 h at RT with TBS Blotto A (Santa Cruz Biotechnology, Dallas, TX, USA) and exposed to primary antibodies diluted in TBS Blotto A overnight at 4 °C with agitation. Membranes were subsequently washed three times with 1X-Tris buffered Saline with Tween-20 (TBS-T) (Santa Cruz Biotechnology) and exposed to the horseradish peroxidase (HRP)-conjugated secondary antibody diluted in 1X-TBS-T for 1 h. Membranes were then washed three times in 1X-TBS, developed with an enhanced chemiluminescence (ECL) reagent (Cell Signaling Technology, Danvers, MA, USA), and exposed to the ChemiDoc™ MP Imaging System to detect the chemiluminescence signals (Bio-Rad Laboratories). The primary antibodies used were rabbit anti-CDC6 (ab109315 at 1:2000, Abcam, Cambridge, UK), rabbit anti-CDT1 (ab202067 at 1:2000, Abcam), rabbit anti-GMNN (ab246509 at 1:1000, Abcam), and mouse anti-ß-Actin (5441, at 1:5000, Sigma-Aldrich, St. Louis, MO, USA) as a loading control.

## 3. Results

### 3.1. Rereplication Analysis by Molecular Combing

HSkM cells previously shown to harbor gene amplifications during their differentiation towards myotubes were differentiated according to the manufacturer’s instructions and were subjected to fiber combing. We modified the protocol for the labeling procedure published by Neelsen KJ and coworkers to detect the rereplication events [15]. Pulse treatments with successive thymidine analogue supplementations (EdU/CldU and/or IdU/CldU, and CldU/IdU) were used to detect replication restarts. Replication events were labelled during first analogue pulse and were again labelled with a second analogue during the replication restart. Both analogue combinations and a reversed analogue order revealed similar results and were confirmed in independent biological replicates of the experiments. 

In brief, cells were differentiated, and pulse treated with the thymidine analogue EdU for 90 min. After the PBS wash step and 15 min incubation in fresh media without the thymidine analogue, the cells were pulse treated with the thymidine analogue CldU for 45 min. We selected EdU and CldU to exclude cross-reaction between antibodies, thymidine analogues, and unspecific Click-iT reactions. The incorporation of EdU was recognized with an Alexa fluorescence labeling Click-iT reaction, and the incorporation of CldU was recognized using the rat-anti BrdU antibody, followed by the fluorescence labeled anti-rat secondary antibody. The clear discrimination of both EdU and CldU is demonstrated in Figure 1A,B. Cells treated with EdU for 1.5 h alone and subjected to the combined EdU and CldU detection procedures revealed only EdU incorporation, and cells treated with CldU for 1.5 h alone, subjected to the detection procedure for both EdU and CldU, revealed only CldU incorporation. In addition, HSkM cells were pulse treated with IdU/CldU and analyzed by EasyComb (Genomic Vision) to monitor for possible adverse effects of EdU on cells.

From previous experiments, it was known that gene amplification was present during the first 24 h after differentiation induction. Thus, we expected rereplication to be the cause for gene amplifications occurring a short time after differentiation induction. To determine the time window during which rereplication occurs, we started the pulse treatment at 8 different time points, i.e., at every hour after differentiation induction, as shown in the schematic in Figure 2A. Experiments were done in two biological replicates. Following the pulse treatment, cells were subjected to fiber combing using the molecular combing system of Genomic Vision. The DNA was combed with a constant stretching factor of 2 kb per µm. Rereplication was detected as simultaneous EdU and CldU incorporation at timepoints between 2 h and 6 h after the differentiation start. No rereplication was detectable at time points 0 h, 1 h, and 7 h after the differentiation start. In conclusion, rereplication occurred in a restricted time window during early differentiation. Representative results are presented in Figure 2B,C.

To determine the length and frequency of incorporated thymidine analogues in fibers of HSkM cells, experiments were repeated with IdU/CldU pulses at 1 h, 4 h, and 6 h after the differentiation induction. As a control, HSkM cells that were allowed to proliferate were pulsed with EdU/CldU 4 h after starting the experiments, as shown in Figure 3A. After 4 h and 6 h of differentiation, many fibers showed a simultaneous incorporation of IdU and CldU, which are visible as yellow stained fiber stretches indicative of a replication restart. A representative overview for rereplication, after 4 h of differentiation, is shown in Figure 3B. Additional YOYO DNA counterstains, in blue, are shown in the upper part of Figure 3B. Representative fibers of the experiment shown in Figure 3A were displayed as enlarged views in Figure 3C. Rereplication was confirmed to occur at 1 h, 4 h and 6 h after the differentiation start, whereas rereplication was not detected in proliferating cells. For quantitative analyses, we determined the frequency of red-, green-, or yellow-containing fibers, compared to blue fibers, with no analogue incorporation. Proliferating HSkM cells revealed no yellow fibers, indicating no rereplication. During the differentiation, the frequency of yellow-containing fibers, indicative of rereplication, increased from 2.3% after 1 h of differentiation, to 7% after 4 h, and 6% after 6 h of differentiation. Many fibers contained up to four discrete yellow traces. Results are summarized in Table 1. 

We further determined the length of the incorporated thymidine analogues in fibers of HSkM cells that were proliferated for 4 h or differentiated for either 1 h, 4 h, and 6 h, as displayed in the box-and-whisker plots in Figure 4. In proliferating HSkM cells, the lengths of red and green traces were similar to those of differentiating cells. Yellow traces, indicative of rereplication, were not detected. In differentiating cells, yellow fibers were detectable after 1 h, 4 h, and 6 h of the differentiation induction. The yellow fibers, after 1 h, were mostly 10 kb or less in length. In contrast, after 4 h and 6 h of differentiation induction, the yellow traces showed an increase in length.

### 3.2. EdU Incorporation and FISH Analysis

In order to investigate rereplication as a source of gene amplification, the EdU treatment was carried out during the timeframe with the highest frequency of rereplication detected by molecular combing, i.e., between 4 h and 8 h after the differentiation induction. Experiments were done in four (differentiation) and two (proliferation) biological replicates. In detail, we pulse treated HSKM cells 3 h after differentiation induction with a 3 h EdU-pulse, and 5 h after differentiation induction with a 45 min EdU-pulse (Figure 5A,B). Next, cells were allowed to differentiate for 24 h and were fixed for subsequent fluorescence in an in situ hybridization using a BAC probe for *MDM2. MDM2* was previously shown to be amplified in HSkM cells after 24 h of differentiation using FISH and qPCR [7]. We pulse treated proliferating HSkM cells 3 h after the beginning of cultivation with a 3 h EdU-pulse (Figure 5C). Next, cells were allowed to proliferate for 24 h and, subsequently, were fixed for fluorescence in an in situ hybridization. HSKM cells revealed a simultaneous EdU stain and a *MDM2* gene amplification during differentiation, as shown by representative images in Figure 5. We analyzed 252 nuclei from differentiating HSkM cells that were treated with a 3 h long EdU-pulse given 3 h after the begin of differentiation. Out of the total number of nuclei, 56% revealed two signals for *MDM2* without EdU staining, 21% revealed two signals for *MDM2* with EdU staining, 13% revealed MDM2 amplification with EdU staining, and 10% revealed *MDM2* amplification without EdU staining. Representative results are shown in Figure 5B. We analyzed 25 nuclei from proliferating HSkM cells that were treated with a 3 h long EdU-pulse given 3 h after the beginning of the proliferation. Out of the total number of nuclei, 84% of the nuclei revealed two signals for the *MDM2* gene without EdU staining, 12% revealed two signals of *MDM2* with EdU staining, and only 4% revealed *MDM2* amplification with EdU staining. Simultaneous EdU staining and *MDM2* amplification during differentiation indicates rereplication as a possible source of gene amplification. Representative results are shown in Figure 5C. Results are summarized in Table 2.

### 3.3. Expression Analysis of Replication Relevant Genes Using RT-qPCR and Western Blot

In normal human cells, rereplication is prevented by a restricted replication induction which allows for only one replication per cell division. Several genes have been identified that are essential to restricting replication [16]. CDT1 and CDC6 are essential for origin licensing and the initiation of replication. The overexpression of these genes can lead to rereplication in primary cells [14]. An expression analysis was done in two biological replicates. Here, we analyzed *CDT1* and *CDC6* gene expressions following a differentiation induction in HSkM cells. We detected a 1.7-fold increase in *CDC6* gene expression 3 h after differentiation induction. Elevated *CDC6* gene expression levels, as compared to the starting level at 0 h, were found up until 12 h after differentiation induction. Reduced *CDC6* levels below the starting expression level were found at 24 h. *CDT1* showed a 1.2-fold increase, 3 h after differentiation induction, followed by lower expression levels. The *GMNN* expression showed an alternating expression level with an increase after 3 h, a decrease after 6 h, and an increase again after 9 h and 12 h of the differentiation induction. An increase in *CDC6* and *CDT1* mRNA expressions, followed by a decrease in *GMNN* mRNA expression, likely enables rereplication. The results of the RT-qPCR (reverse transcription-quantitative polymerase chain reaction) analysis are shown in Figure 6A.

Next, we examined the protein expression of CDT1, CDC6, and GMNN during the time window between 3 h and 7 h, during which mRNA expression changes were most prominent. The CDT1 expression was strong from the beginning until 5 h of differentiation, and subsequently decreased for the remaining time. In contrast, the CDC6 expression increased from the beginning until after 7 h of differentiation, and subsequently decreased with no CDC6 expression detectable at 24 h. The geminin (GMNN) expression was strong during the first 3 h of differentiation, and decreased afterwards to a barely detectable expression level. The strong protein expression of CDT1 concomitant with an increase in CDC6 and a strong decrease in GMNN enabled the requirements for a replication restart and rereplication during early differentiation. The results of the Western blot analysis are summarized in Figure 6B.

## 4. Discussion

Replication normally occurs once per cell cycle [17]. As demonstrated in Drosophila, this restriction can be circumvented to enable endoreplication during development. It is also known that during development, various cell types reveal polyploidy, including mammalian trophoblasts, mammalian megakaryocytes, and plant cells. Endocycles and endoreplications are postulated to account for this copy number gain [18]. Several key regulators of replication control were reported to allow for rereplication when experimentally over-expressed [14]. It has been reported that *CDT1* over-expression resulted in rereplication and gene amplification. *CDT1* is degraded during the mitotic cell cycle by E3 ubiquitinylation to prevent rereplication. Likewise, *CDC6* is degraded to prevent rereplication. In contrast, the repression or deletion of geminin (GMNN), an important regulator of replication control, resulted in rereplication [19].

Studies by Munoz and colleagues revealed that *CDC6* overexpression was the limiting factor for origin relicensing, but it was not sufficient for rereplication induction. Likewise, *CDT1* overexpression alone was not sufficient for relicensing and rereplication [14]. Only the simultaneous overexpression of *CDC6* and *CDT1* led to rereplication. Our expression analysis revealed both an increase in both *CDC6* and of *CDT1* expressions at 3 h after differentiation induction in HSkM cells. Protein expression levels of CDC6 and CDT1 were also elevated at specific time points. These observations are consistent with the idea that altered expression levels of CDC6 and CDT1 enable the observed rereplication in HSkM cells.

In addition, between 3 h and 6 h after differentiation induction, *GMNN* expression decreased. Accordingly, GMNN protein expression was reduced after 4 h and was not re-elevated to the initial expression level. These findings are consistent with the hypothesis that reduced or depleted GMNN expression contributes to the rereplication during this time window. 

Although fiber combing is a very powerful technique for the visualization of rereplication, it requires rigid controls to exclude cross-detection between various thymidine analogues. Cross-detection may be due to antibody cross detection, fiber clumping, or label persistence. 

As for antibody cross-detection, we used Edu and CldU as thymidine analogues that were identified with different detection systems, i.e., EdU was detected by a chemical Click-iT reaction, and CldU by an antibody binding. Accordingly, we did not find evidence for a cross detection between the thymidine analogues EdU and CldU. To further monitor repeated restarts in a given DNA sequence, we included a third thymidine analogue (IdU) pulse. Due to cross-detection of EdU and IdU, it was, however, not possible to use all three analogues (EdU, IdU, and CldU) in one experiment.

To monitor fiber clumping as a further potential cause of cross-detection, we counterstained the DNA with YOYO and ensured that multiple fibers were not overlapping. 

To counteract the potential persistence of labeling, we employed a PBS washing step after the first analogue pulse. The PBS washing step resulted in a reduced yellow staining, indicating a reduced analogue persistence due to the addition of a washing step. A further reduction in yellow staining was found after an additional incubation with an analogue free medium for 15 min. Nevertheless, we cannot completely rule out label persistence after the first pulse with the thymidine analogue. Towards a discrimination between rereplication and the potential label persistence, we only considered long uninterrupted yellow fibers. As a threshold, we chose a length of >10 kb. 

Preliminary observations during our rereplication fiber-combing analysis, during differentiation, may indicate asymmetric replication bubbles (data not shown). Up until now, rereplication was mostly reported as a symmetric process described as an onion-like-shaped symmetric replication bubble. The *Drosophila* chorion gene amplification, with symmetric replication bubbles, were shown as early as 1984 by Stark and Wahl [20] and, more recently, in 2017 by Hua and Orr-Weaver [21]. However, electron microscopy studies by Osheim 1988 suggested a lack of symmetry of the replication bubbles and replication forks during *Drosophila melanogaster* chorion gene amplification [11]. Further investigations with higher resolutions are required to clarify this issue.

To correlate rereplication and gene amplification, we pulse treated differentiating HSkM cells with EdU for 3 h during the time window that showed a high rereplication frequency by fiber combing. We evidenced *MDM2* gene amplification by fluorescence in in situ hybridization. Multiple fluorescence signals for *MDM2,* indicative of *MDM2* gene amplification, were colocalized with EdU incorporation in 13% of analyzed cell nuclei. In addition, 10% of analyzed nuclei revealed *MDM2* amplification without EdU incorporation. This observation is likely due to the amplification after EdU supplementation. Out of the analyzed nuclei, 21% revealed EdU incorporation with a normal diploid copy number of *MDM2.* Previous amplification analyses revealed a heterogeneous pattern of gene amplifications during myogenic or neural differentiation [6,7], and it is very likely that other genes adjacent to *MDM2* were amplified during EdU incorporation. As a control-experiment, we pulse treated proliferating HSkM cells with EdU for 3 h during the same time window. The vast majority of nuclei revealed diploid *MDM2* fluorescence signals and no EdU incorporation (84%). Only 12% of nuclei revealed Edu incorporation with diploid *MDM2* fluorescence signals and only one nucleus (4%) revealed more than two *MDM2* fluorescence signals with EdU incorporation. 

In conclusion, our study is the first to report rereplication in normal human cells during a restricted time window of early differentiation. During this short time window, we also found deregulated expression levels of genes involved in replication control. It remains to be investigated whether this time restriction is an essential component of controlled gene amplification during the differentiation processes in mammals. Additionally, it remains to be clarified as to what extent a temporal and spatial suspension of the time- and cell type-specific rereplication restriction contributes to uncontrolled gene amplifications during tumor development.

## Figures and Tables

**Figure 1 cells-11-01060-f001:**
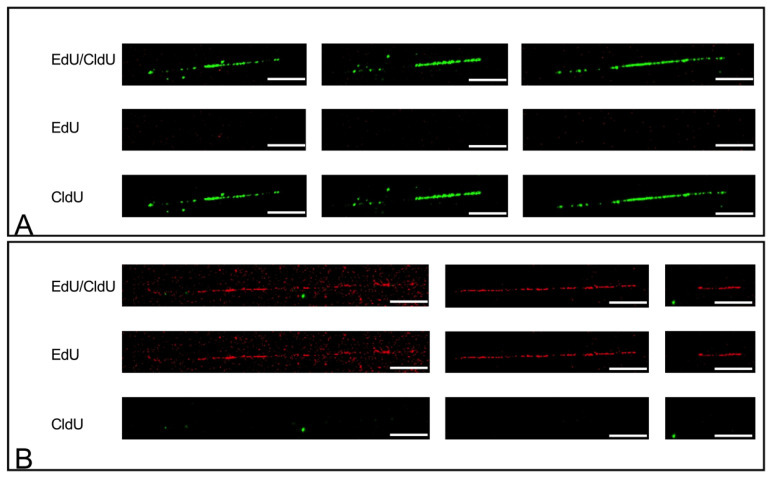
Exclusion of cross detection between thymidine analogues EdU and CldU. HSkM cells were allowed to differentiate for 4 h and subsequently pulse treated with CldU (**A**) or EdU (**B**) for 90 min. Cells were harvested and used for fiber combing. Fibers were sequentially analyzed by two detection procedures: Edu Click-iT reaction-Alexa594 (red) followed by immunofluorescence using rat-anti-BrdU primary antibody and Alexa-488 (green) coupled secondary antibody for CldU incorporation. Both control experiments revealed no cross reaction of Edu Click-iT and immunofluorescence detection of CldU as shown in (**A**) with only green fibers in cells with CldU pulse-treatment and in (**B**) with only red fibers in cells with EdU pulse-treatment. Scale bars represent 10 µm.

**Figure 2 cells-11-01060-f002:**
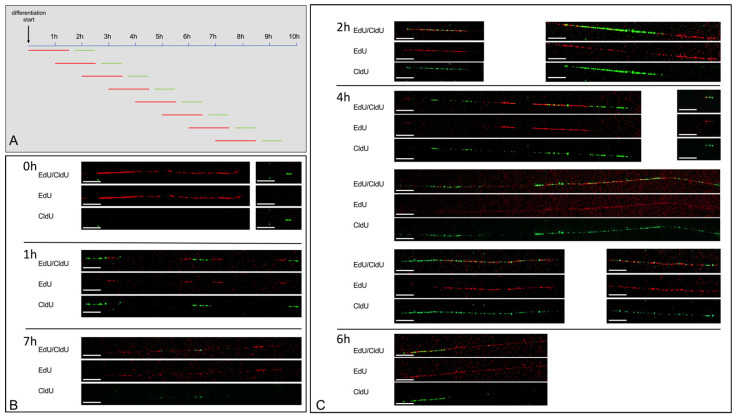
Rereplication analysis. EdU/CldU pulse treatment experiments started at 8 different time points, i.e., at time points 0 h, 1 h, 2 h, 3 h, 4 h, 5 h, 6 h, and 7 h after differentiation induction of HSkM cells. An overview of pulse time duration for EdU and CldU and the period in between both treatments is given in (**A**). Results of fiber analysis for time points 0 h, 1 h, and 7 h revealed no rereplication as represented in (**B**). Results of fiber analysis for time points 2 h, 4 h, and 6 h revealed rereplication as represented in (**C**). The fibers are shown both as a merged view for EdU/CldU and as single views for either EdU or CldU. Scale bars represent 10 µm.

**Figure 3 cells-11-01060-f003:**
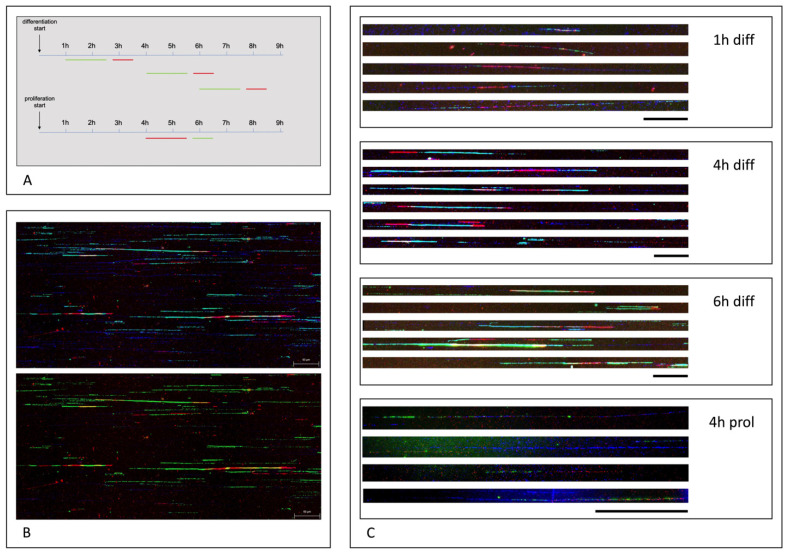
Rereplication analysis. EdU/CldU pulse treatment experiments started 4 h after proliferation induction of HSkM cells. IdU/CldU pulse treatment started 1 h, 4 h, and 6 h after differentiation induction of HSkM cells. An overview on pulse time duration and the period between both treatments is given in (**A**). Representative scan images of analyzed fibers including fibers with and without YOYO stain (upper and lower part of the figure) is given in (**B**). Fiber analysis of differentiation at time points 1 h, 4 h, and 6 h showed rereplication as represented in (**C**). Fiber analysis of proliferation at time point 4 h showed no rereplication. Scale bars represent 50 µm = 100 kb. YOYO counterstain of DNA fibers is visualized in blue color.

**Figure 4 cells-11-01060-f004:**
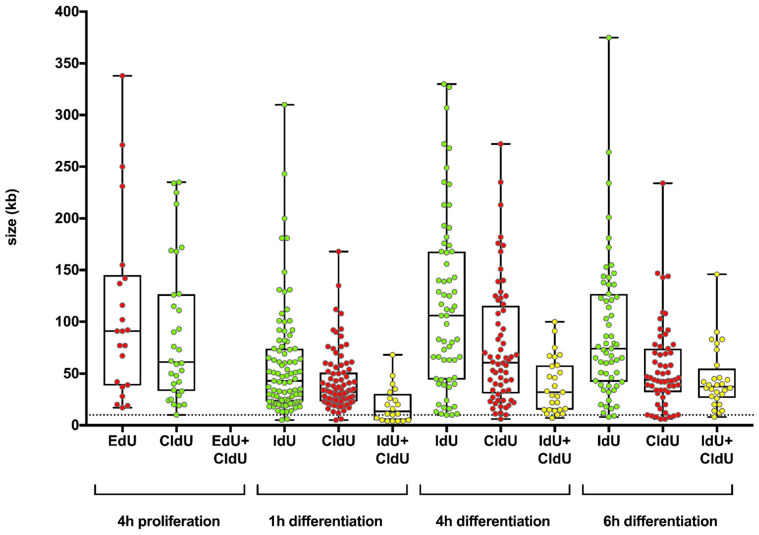
Length and distribution of thymidine analogue integration during proliferation and differentiation. Fibers of HSkM cells, which proliferated for 4 h or differentiated for 1 h, 4 h, or 6 h, were analyzed for integrated thymidine analogues and the length of uninterrupted thymidine integration (dot). The boxes indicate the 2nd and 3rd quartiles, and the whiskers represent the minimum and maximum values. The EdU fiber lengths are shown in red, the CldU fiber lengths are shown either in green or in red, the IdU fiber lengths are shown in green, and the simultaneous IdU and CldU fiber lengths are shown in yellow. Dotted lines represent a threshold of 10 kb. The Y axis shows the length of the colored fibers, and the X axis shows the thymidine analogues used in the different experiments.

**Figure 5 cells-11-01060-f005:**
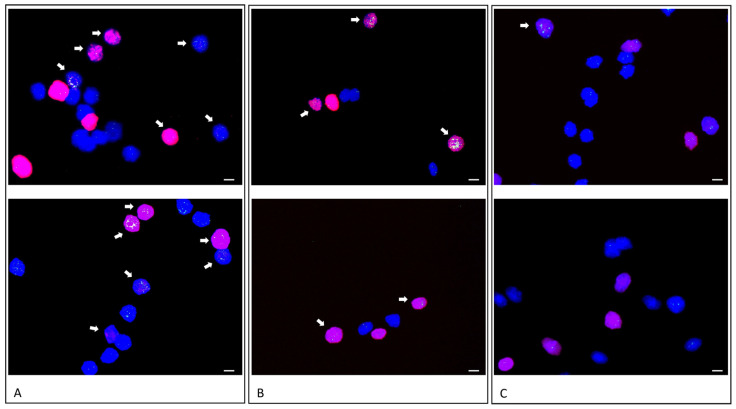
*MDM2* gene amplification in EdU positive HSkM cells. Differentiating HSkM cells were EdU pulsed for 45 min, 5 h after differentiation induction (**A**). Differentiating HSkM cells were pulsed for 3 h with EdU, 3 h after differentiation induction (**B**). Proliferating HSkM cells were pulsed for 3 h with EdU, 3 h after the proliferation start (**C**). After EdU treatment, all cells were washed twice with PBS and supplied with fresh medium for additional 18 h. EdU incorporation was detected using Edu Click-iT reaction with Alexa-Fluor 594 (red). Fluorescence in situ hybridization of BAC probe RP11-611O2 (*MDM2*) is shown as a bright green labeled fluorescence signal with DAPI counterstain of nuclei in blue fluorescence. Arrows point to nuclei with *MDM2* amplification and EdU incorporation. The upper and the lower panel are two exemplary pictures for each experiment (**A**–**C**). Scale bars represent 10 µm.

**Figure 6 cells-11-01060-f006:**
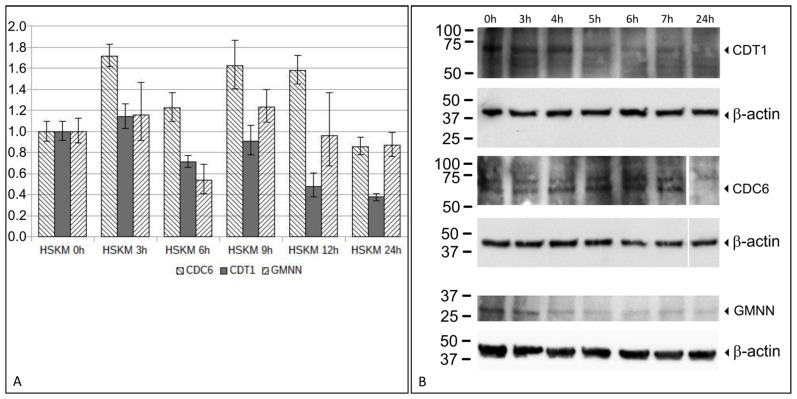
Gene and protein expression analysis of replication regulating genes *CDC6*, *CDT1,* and *GMNN*. HSkM cells were differentiated for 24 h and RNA was isolated at the indicated time points and used for RT-qPCR (**A**). QuantiTect Primer Assays for *CDT1*, *CDC6,* and *GMNN* were used as test genes and compared to *TBP* and *GAPDH* housekeeping genes. RT-qPCRs were done in two replicates and standard deviation is indicated as a vertical line. HSkM cells were differentiated for 24 h and protein was isolated at the indicated time points and used for Western blot analysis (**B**). Primary antibodies against CDT1, CDC6, and GMNN were probed in separate blots and each blot was sequentially probed with ß-actin as loading control. Protein sizes were indicated and revealed expected protein sizes for all tested antibodies.

**Table 1 cells-11-01060-t001:** Frequency of thymidine analogue incorporation (colored fiber traces).

	Fibers n	Yellow n (%)	Red n (%)	Green n (%)	Red and Green n (%)
**4 h proliferation**	520	0 (0)	9 (1.7%)	15 (2.9%)	17 (3.3%)
**1 h differentiation**	221	5 (2.3%)	3 (1.4%)	3 (1.4%)	3 (1.4%)
**4 h differentiation**	196	14 (7%)	11 (5.6%)	51 (26%)	17 (8.7%)
**6 h differentiation**	182	11 (6%)	6 (3.3%)	94 (52%)	19 (10%)

**Table 2 cells-11-01060-t002:** Analysis of *MDM2* copy number and EdU incorporation.

	3–6 h Differentiation	3–6 h Proliferation
**Number of nuclei**	252	25
**2 *MDM2* signals and no EdU stain**	199 (56%)	21 (84%)
**2 *MDM2* signals and EdU stain**	74 (21%)	3 (12%)
**>2 *MDM2* signals and EdU stain**	45 (13%)	1 (4%)
**>2 *MDM2* signals and no EdU stain**	34 (10%)	0 (0%)

## Data Availability

All relevant data are within the paper.

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
