# Peer review of "A Temporary Pause in the Replication Licensing Restriction Leads to Rereplication during Early Human Cell Differentiation"

_cells, 2022, doi:10.3390/cells11061060_

Round 1
Reviewer 1 Report
Title: Temporary pause of replication licensing restriction leads to rereplication during early differentiation
In this paper Minet el al. investigate whether rereplication is a possible mechanism leading to gene amplifications in human myoblasts following differentiation induction. The question is relevant and interesting. However, I still find several of the the experimental results to be below the standards required in the field.
The methodology used in the paper seems appropriate in general. However, it is still not clear how many replicas were analyzed (e.g. Figures 2, 3, 4 and 5 and Tables 1 and 2).
The authors have now included a “4h proliferation” control with cells cultured in non-differentiating medium.
Major points:
- DNA molecular combing experiments. In Figure 2 and 3 and Table 1 the authors present representative images of combed fibers for each time point to detect rereplication after differentiation induction and quantification of the frequency of re-replication. It is still not clear how many independent experiments have been performed. Although the percentage of the scored fibers that presented rereplication is given in Table 1 it looks like a low number of fibers have been scored per condition, especially when most of those fibers were not labelled. For example, at 4h differentiation less than 50 % of fibers contained any labelled analogue, which means than less than 100 labelled fibers were scored coming from (as far as it is stated in the text) maybe only one experiment. Given that has been proved that sample size is critical in DNA fiber analysis (Techer et al, J. Mol. Biol. (2013) 425, 4845–4855) I find it important to increase the number of scored fibers. It is also incorrect to compare conditions with such a different number of labelled fibers (41, 14, 93 and 130) because the differences found in yellow tracks frequency could in part be due to the different number of fibers analysed. Even though the results seem clear, they cannot be accepted as real until reproducibility of them is proven by repeated independent experiments with enough number of measurements per experiment.
On the other hand, in Figure 4 the authors show the length of labelled tracks found in the combing experiments. In this figure, it is not clear to me which tracks are the authors measuring, as the number of dots in each condition in the graph seems higher than the number stated in table one (e.g. in 4h differentiation in the graph I count 27 yellow dots, while in the table only 14 yellow tracks were scored). Even more, as previously mentioned, extracting information from such low number of fibers seems incorrect. I do not think this data adds anything to the story and should be removed from the manuscript.
- FISH analysis. In Figure 5, the red (EdU) staining seems over-exposed, masking the green signal. Please amend this point.
In Table 2, same as in the combing experiments, it is not clear how many independent experiments were performed. Also, in the proliferation control only 25 nuclei, presumably from a single experiment, were scored. That is only 10% of the number of nuclei scored for the differentiation condition. Please, increase that number of experiments and nuclei so both conditions are comparable.
- Expression analysis of CDC6, CDT1 and GMNN. Again, it is not stated how many independent experiments were performed for obtaining these results. In the case of the immunoblot, detection of the proteins in the chromatin bound fraction would be more informative than the levels in whole cell extracts. Nevertheless, total protein levels go in line with the authors interpretation.
Minor points:
The minor points have been addressed.
Author Response
DNA molecular combing experiments. In Figure 2 and 3 and Table 1 the authors present representative images of combed fibers for each time point to detect rereplication after differentiation induction and quantification of the frequency of re-replication. It is still not clear how many independent experiments have been performed. Although the percentage of the scored fibers that presented rereplication is given in Table 1 it looks like a low number of fibers have been scored per condition, especially when most of those fibers were not labelled. For example, at 4h differentiation less than 50 % of fibers contained any labelled analogue, which means than less than 100 labelled fibers were scored coming from (as far as it is stated in the text) maybe only one experiment. Given that has been proved that sample size is critical in DNA fiber analysis (Techer et al, J. Mol. Biol. (2013) 425, 4845–4855) I find it important to increase the number of scored fibers. It is also incorrect to compare conditions with such a different number of labelled fibers (41, 14, 93 and 130) because the differences found in yellow tracks frequency could in part be due to the different number of fibers analysed. Even though the results seem clear, they cannot be accepted as real until reproducibility of them is proven by repeated independent experiments with enough number of measurements per experiment.
In the revised version we now clarify that we performed two independent experiments for each of the 7 differentiation times to identify the time window with the highest frequency of rereplications. We also clarify that we performed additional experiments with EdU/CldU, IdU/CldU and EdU/CldU/IdU. These additional experiments always detected rereplication. By contrast, our three independent experiments with undifferentiated cells did not detect rereplication for any of the different time points. We also made clear that the focus of our paper was on the identification of rereplication after differentiation induction and not on a quantification of the replication events.
On the other hand, in Figure 4 the authors show the length of labelled tracks found in the combing experiments. In this figure, it is not clear to me which tracks are the authors measuring, as the number of dots in each condition in the graph seems higher than the number stated in table one (e.g. in 4h differentiation in the graph I count 27 yellow dots, while in the table only 14 yellow tracks were scored). Even more, as previously mentioned, extracting information from such low number of fibers seems incorrect. I do not think this data adds anything to the story and should be removed from the manuscript.
We apologize for not being clear on this issue. We now clarify that the table displays the number of fibers that show thymidine analogue incorporation. By contrast, the figure displays the overall number of colored traces. Due to the fact that many fibers show multiple thymidine analogue traces the number of these fibers can be lower than the overall number of thymidine analogue incorporations. This has now been made clear in the manuscript.
FISH analysis. In Figure 5, the red (EdU) staining seems over-exposed, masking the green signal. Please amend this point.
We do not agree with the reviewer on this point. In several replicates we identified nuclei that show a very bright signal for EdU next to nuclei with a rather moderate signal for EdU. This argues strongly against an over-exposition.
In Table 2, same as in the combing experiments, it is not clear how many independent experiments were performed. Also, in the proliferation control only 25 nuclei, presumably from a single experiment, were scored. That is only 10% of the number of nuclei scored for the differentiation condition. Please, increase that number of experiments and nuclei so both conditions are comparable.
We now clarified that we performed four independent experiments for the analysis of the differentiating HSkM cells. Two independent experiments were performed for the proliferating HSkM cells that showed a very low number of EdU stained nuclei.
Expression analysis of CDC6, CDT1 and GMNN. Again, it is not stated how many independent experiments were performed for obtaining these results. In the case of the immunoblot, detection of the proteins in the chromatin bound fraction would be more informative than the levels in whole cell extracts. Nevertheless, total protein levels go in line with the authors interpretation.
We now state that the expression analysis was performed in two independent biological replicates.
Reviewer 2 Report
In the manuscript Cells-1623796, Minet et al investigated rereplication events during early myotube differentiation of human skeletal myoblast cells, using fiber combing and combinatorial pulse-treatments with EdU/CldU and IdU/CldU. The authors found rereplication occurred during a restricted time window between 2 h and 8 h after differentiation induction. Specifically, the authors also observed concurrent MDM2 gene amplication in cells that rereplication was detected.
Specific concerns,
- It will be helpful if the authors in the main text can explain why combinatorial pulse-treatments with EdU/CldU and IdU/CldU is able to detect the rereplication events, i.e., which respective event each of them can detect.
- It will be helpful if the authors can clearly state in each subsection of the Result Section in the main text what are the respective results and conclusions from the Figures.
- Figure 5: 1) bar scales are missing; 2) what are the lower and upper panels in the A, B, and C panels? 3) Where is the green color? 4) It seems that there are also nuclei with MDM2 amplification and EdU incorporation in the lower panel of C.
- Citations 7 and 16 are the same.
Author Response
Specific concerns,
It will be helpful if the authors in the main text can explain why combinatorial pulse-treatments with EdU/CldU and IdU/CldU is able to detect the rereplication events, i.e., which respective event each of them can detect.
We now explain why combinatorial pulse-treatments detects rereplication events, i.e. we explain what event is detected by what combination.
It will be helpful if the authors can clearly state in each subsection of the Result Section in the main text what are the respective results and conclusions from the Figures.
In the result section we now specifically refer to the Figures and the conclusion that can be drawn from the results shown.
Figure 5: 1) bar scales are missing;
We now included scale bars for 10µm in Figure 5.
What are the lower and upper panels in the A, B, and C panels?
We now clarify in the figure legend that the upper and the lower panel represent two exemplary figures.
Where is the green color?
We apologize for not been precise on this description. The color is now referred to as bright green.
It seems that there are also nuclei with MDM2 amplification and EdU incorporation in the lower panel of C.
We were very careful not to overinterpret the data. The according signals are too weak and too small to indicate MDM2 amplification in this image.
Citations 7 and 16 are the same.
We apologize for this error. In the revised version the citation 16 is replaced by citation 7.
Moderate English changes required
Manuscript was now revised by a native speaker.
This manuscript is a resubmission of an earlier submission. The following is a list of the peer review reports and author responses from that submission.
Round 1
Reviewer 1 Report
Title: Temporary pause of replication licensing restriction leads to rereplication during early differentiation
In this paper Minet el al. investigate whether rereplication is a possible mechanism leading to gene amplifications in human myoblasts following differentiation induction. The question is relevant and interesting. However, I find several of the the experimental results to be below the standards required in the field, making it difficult to draw conclusions at this stage of the study.
The methodology used in the paper seems appropriate in general. However, the results presented rely on representative images and it is not clear how many cells or replicas were analyzed (e.g. Figures 2 and 3). In their present form, it would be difficult to extract any conclusion from the data.
Also, it would be relevant to know what happens in cells in which differentiation is not induced and/or in fully differentiated cells. Similarly, it would be pertinent to compare these results to experiments done in a system where no gene amplification occurs during differentiation.
Major points:
- DNA molecular combing experiments. In Figure 2 the authors present representative images of combed fibers for each time point to detect rereplication after differentiation induction. While the images could be convincing (although they should be improved to better show the signal overlap), conclusions cannot be made without appropriate quantification: how many experiments have been performed? how many fibers were scored per experiment? what percentage of the scored fibers presented rereplication? Also, it seem essential to know the level of rereplication in control conditions (no differentiation induction) and/or in cells that do not undergo gene amplification.
- FISH analysis. Although the image in Figure 3 showing MDM2 gene amplification seems clear, I do not agree with the sentence: “HSKM cells revealed simultaneous EdU incorporation and MDM2 gene amplification ….” (lines 201-202). I guess that the point is to show that gene amplification is dependent on DNA synthesis. In the image presented only 3 of the 6 cells that show strong positive FISH signal are also positive for EdU. This means that either the image is not representative or the sentence is an over-interpretation. Again, a precise quantification of the images is needed (% of EdU+, FISH+ and EdU-and-FISH+). Also, controls with a normal cell with no gene amplification would be helpful.
- Expression analysis of CDC6, CDT1 and GMNN. The function of these proteins is exerted when they bind DNA and their regulation is mainly through posttranscriptional mechanisms. Rather than simple mRNA expression levels, the study would benefit from the analysis of the protein levels and their association with chromatin. Also, it is shown in Figure 4 that at 6h after differentiation induction (when rereplication is apparently maximum) CDT1 levels are very low, even if GMNN levels are also very low. I cannot see how such low CDT1 levels would promote rereplication unless these relative mRNA levels are different from the protein levels.
- Rereplication bubble-hypothesis. In Figure 5 authors try to demonstrate the existence of symmetric and asymmetric bubbles of rereplication by means of staining patterns in their combing experiments. However, this seems speculative at this point. It is hard to believe that optical imaging allows for the discrimination of individual DNA fibers to that level of detail, given that resolution of images from this kind of microscope is in the order of 0.25 micrometers while chromatin fibers are around 30 nanometers in diameter.
Minor points:
- DNA molecular combing experiments. Although the authors make a good effort to demonstrate that there is no cross-reaction between EdU and CldU signals in their immnunodetection method (Figure 1), they lack the usual DNA staining in order to account for the integrity of the DNA fibers.
- FISH analysis. If the purpose of the experiment is to show correlation between DNA synthesis (EdU) and gene amplification (FISH signal), the authors should combine those red and green signals in the same overlaid image.
Reviewer 2 Report
The manuscript by Minet et al concerns gene amplification during human cell differentiation. The authors present data that they interpret as evidence for DNA re-replication occurring during early myotube differentiation of human skeletal myoblast cells. However, This reviewer is unconvinced by the data presented - regarding both the evidence for amplification and the underlying mechanism. Much better data - including convincing controls - is required before this work is suitable for publication.
The main evidence provided for re-replication is an overlap in the labelling of DNA strands by EdU and CldU when these precursors are added sequentially to cells and the DNA is then examined by microscopy.
Although the two labels are chemically distinct, the first concern I have is the possibility of a small degree of contamination between the two labels in cells not undergoing re-replication. This point is partially addressed in Figure 1, but this is purely qualitative data and so cannot be used as a control for subsequent experiments and only addresses the degree to which DNA containing one label is detected in the channel for the other label; the degree of overlap due to label persistence is not addressed. Note that once cells have taken up EdU (the first label) it will not be washed out and will only be depleted from the intracellular thymidine pool by ongoing DNA synthesis; depending on labelling intensity, this can take 20 minutes or more to become undetectable. What is required is an experiment where the two labels are added sequentially to cells not expected to undergo re-replication, and then the degree of signal overlap is quantified in a way that would be consistent with the analysis of subsequent experiments. This would show the degree of label cross-reaction and also the extent to which the first label (EdU) is cleared from cells before the second label (CldU) is added.
The fibre images presented in Figure 2 presents examples of the types of fibre patterns recovered, but no quantitative analysis is performed. What is significantly more important is the presentation of statistical information describing different classification of fibre types, their lengths and the fraction of red:green:yellow found across the time course. This should then be compared to the data obtained in cells not undergoing re-replication as described above. To this reviewer's eye there is a low frequency of long double stained yellow fibres and a high proportion of CldU staining in short regions where EdU staining is either reduced of absent. The CldU labelling appears rather patchy and does not clearly show the sort of uniform labelling expected of forks progressing for 10 kb or more as expected of rereplication.
This brings me to another important concern about interpreting the DNA fibre data. When re-replication occurs, a Watson strand labelled with EdU should be base-paired with a complementary Crick strand labelled with CldU. It is not possible by microscopy to be sure that this has happened, rather than a Watson strand being partially replicated and labelled with EdU and then at a later stage the same Watson strand having its gaps filled in with CldU. The second mechanism (not representing rereplication) would give patchy overlap (yellow staining) of the two labels consistent with the pictures shown. Considering that EdU is not well-tolerated by cells, its incorporation into nascent DNA could potentially lead to replication fork stalling, checkpoint activation or active DNA repair, thereby increasing the degree of double-labelling of the same strand by two labels. It is not trivial to rule out such gap-filling reactions. Meselson-Stahl density substitution is one way of doing this. A less definitive approach would be to restrict analysis to overlaps of long (>10 kb) uninterrupted tracks of both labels which would be consistent with unimpeded fork progression. However, none of the CldU tracks shown in Figure 2 are fully labelled for this sort of length. Counter staining of the fibres with e.g. YOYO, should also be presented to ensure that no fibre clumping has occurred.
Figure 3 attempts to show evidence for re-replication by FISH labelling. Again, only qualitative data is shown and it is not convincing. There is no reason to expect that rereplicated DNA would significantly move its location in the nucleus, and so I would expect FISH probes to label the same number of loci, but the rereplication would labelling would increase the FISH intensity of the loci. This is not what is seen in Figure 3, where some cells show very strong dispersed staining in the nucleus that looks more like an artefact than rereplication. There is no clear correlation between the presence of EdU and the MDM2 detection. These experiments need to include quantitative data about the intensity and distribution of Edu and MDM2 detection, and there should be a comparison between rereplicating cells and normal cells.
Without accompanying cell cycle status data, the data presented in figure 4 is hard to interpret. It should be noted that only 2 replicates have been performed and that a significant number of error bars overlap with one another which makes it difficult to draw any conclusions. More importantly, it should be noted that these proteins are regulated most stringently by cell cycle-dependent protein degradation, and so measurements of protein quantity are they key observation – differences in mRNA levels may signify very little in terms of capability for rereplication.